# Brain Tumor Segmentation Based on Deep Learning’s Feature Representation

**DOI:** 10.3390/jimaging7120269

**Published:** 2021-12-08

**Authors:** Ilyasse Aboussaleh, Jamal Riffi, Adnane Mohamed Mahraz, Hamid Tairi

**Affiliations:** LISAC Laboratory, Department of Computer Science, Faculty of Sciences Dhar El Mahraz, University Sidi Mohamed Ben Abdellah, Fez 30000, Morocco; riffi.jamal@gmail.com (J.R.); adnane_1@yahoo.fr (A.M.M.); hamid.tairi@usmba.ac.ma (H.T.)

**Keywords:** magnetic resonance imaging, brain tumor, segmentation, deep learning, convolution neural networks

## Abstract

Brain tumor is considered as one of the most serious causes of death in the world. Thus, it is very important to detect it as early as possible. In order to predict and segment the tumor, many approaches have been proposed. However, they suffer from different problems such as the necessity of the intervention of a specialist, the long required run-time and the choice of the appropriate feature extractor. To address these issues, we proposed an approach based on convolution neural network architecture aiming at predicting and segmenting simultaneously a cerebral tumor. The proposal was divided into two phases. Firstly, aiming at avoiding the use of the labeled image that implies a subject intervention of the specialist, we used a simple binary annotation that reflects the existence of the tumor or not. Secondly, the prepared image data were fed into our deep learning model in which the final classification was obtained; if the classification indicated the existence of the tumor, the brain tumor was segmented based on the feature representations generated by the convolutional neural network architectures. The proposed method was trained on the BraTS 2017 dataset with different types of gliomas. The achieved results show the performance of the proposed approach in terms of accuracy, precision, recall and Dice similarity coefficient. Our model showed an accuracy of 91% in tumor classification and a Dice similarity coefficient of 82.35% in tumor segmentation.

## 1. Introduction

Brain tumor is one of the most dangerous cancers in the world. It appears when a certain type of brain cell, known as malignant, begins to grow out of control. In the past 30 years, the number of patients diagnosed with brain cancer has significantly increased, affecting many people throughout the world. This increase leads to a high risk of mortality with 241.037 cases in 2018. In 2012, eight million people died from cancer, taking all types of cancer together, while in 2013, six million people worldwide died of brain tumor. 

It is very important to diagnose brain cancer at an earlier stage as it allows for therapy and enhances the rate of survival. Brain tumor treatment options depend on the location, type and size of the tumor and may involve radiotherapy, surgery, chemotherapy or a combination of these options. Medical imaging is used to verify the presence and show certain characteristics of different types of brain tumors. There is a multitude of medical imaging modalities, including magnetic resonance imaging (MRI) and computerized tomography (CT), which are the most common ones used to explore brain cancer. 

The efficient classification and segmentation of tumors from surrounding brain tissues is a crucial task. In fact, an essential step is to exclude normal tissues by segmentation and extract more relevant characteristics of lesions for a better diagnosis. However, segmentation is a difficult task due to the wide variations in size, texture, shape and location of brain lesions. 

For clinical diagnosis, appropriate classification and segmentation of medical images are necessary. Therefore, several algorithms and methods have been presented for manual, semi- and fully automated tumor segmentation due to the complicated tumor segmentation process in the MRI image. Then, manual segmentation is performed by a radiologist which is considered the gold standard. However, expert segmentation is not very precise and is subject to inter-observer variability. Expert segmentation is time-consuming as it involves visualizing spatial and temporal profiles and thus examining many enhanced datasets and pixel profiles while determining the injury boundary. Then the best solution offered by computer vision is to employ fully automated systems using machine learning techniques.

Therefore, numerous machine learning approaches have been applied effectively to recognize a brain tumor. The most popular and well-known supervised classifiers that have been used to classify gliomas were random forests (RFs) and support vector machines (SVMs). Lefkovits et al. [1] built a model using an RF classifier, after extracting features and selecting the pertinent ones, to extract features’ first-order operators (mean, standard deviation, maximum, minimum, median, Sobel, gradient), higher-order operators (Laplacian, difference of Gaussians, entropy, curvatures, kurtosis, skewness), texture features (Gabor filter) and spatial context features. All these features are analyzed to select the importance variable using an appropriate selection of attributes. Szabo et al. [2] proposed a method to segment low-grade gliomas in MRI images; they extracted 104 morphological and Gabor wavelet features, employing an RF as a classifier and neighborhood-based post-processing for output regularization. Zhang et al. [3] presented a method that was divided into three main steps: pre-processing and feature generation (minima, maxima, average, median, gradient, Gabor wavelet features); then the RF was trained to classify normal pixels from positive ones, while post-processing used morphological phase to regularize the shape of detected lesions. Bahadure et al. [4] proposed a method that combined the Berkeley wavelet transform to convert the spatial form into temporal domain frequency and the SVM classifier. Ayachi et al. [5] transformed the segmentation problem into a classification problem; they classified the pixels into normal and abnormal ones based on several features based on intensity and texture, and they employed an SVM such as a classification algorithm. Kwon et al. [6] proposed a spatial probability map for each tissue type, in which all the different tissues in a patient’s brain are segmented. Menze et al. [7] also used spatial regularization with a generative probabilistic model where a healthy brain tumor atlas and a latent brain tumor atlas were combined to segment brain tumors into a series of image sequences. Jayachandran et al. [8] classified MRI images as normal and abnormal in their approach using a fuzzy logic-based hybrid kernel SVM. A classification study of tumors using Gabor wavelet analysis was conducted by Liu et al. [9]; they were used for the extraction of the features, and Gabor filters and an SVM classifier were adopted to classify the tumor.

Deep learning application stands out as an ideal solution since it can extract more prominent features from the whole image than from manually defined features.

The most frequently adopted segmentation approaches based on deep learning require masked images representing the expected result. Certainly, these labels help to guide the learning process in the segmentation task. On the other hand, their preparation remains a time-consuming task, while the expert’s subjectivity presents another problem. To overcome these problems, we proposed a tumor segmentation approach based on CNN architecture without using masked images. Indeed, after predicting the existence of a tumor based on a CNN architecture that we trained without using labels in the form of images but rather in the form of two numbers (0 or 1), we constructed an image from a combination of the gradients of the last layer of features; then we calculated the gradient of each image filter extracted from the last layer and stocked the mean and the maximum of each one into two different vectors, after we multiplied those vectors with all the filters component by component (component1 × filter1, component2 × filter2 ……, component32 × filter32) to obtain the mask, applying a color map and finally post-processing to generate the segmented image.

This article is organized as follows. Related work is provided in Section 2. The proposed method is described in Section 3. The experimental setup is introduced in Section 4. We summarize our results and then discuss them in Section 5. Finally, after the conclusion in Section 6, we present our plans for future research.

## 2. Related Work

Research in the field of tumor segmentation is still active. Recently, deep learning has proven its performance in medical image analysis and retrieval [10,11]. Pixel-based segmentation is a new trend in deep learning methods [12].

The methods cited in this section were divided into different methods that used CNNs. Lyksborg et al. [13] proposed a binary CNN to identify the complete tumor as a cellular automate then smooths the segmentation before a multi-class CNN discriminates the sub-regions of the tumor. Pereira et al. [14] employed an automatic segmentation method when they investigated the use of intensity normalization as a pre-processing step, which, though not common in CNN-based segmentation methods, proved together with data augmentation to be very effective for brain tumor segmentation in MRI images.

In addition, Havaei et al. [15] presented a novel CNN architecture that exploits both local features as well as more global contextual features simultaneously; they explored a cascade architecture in which the output of a basic CNN is treated as an additional source of information for a subsequent CNN. Moreover, Madhupriya et al. [16] used a CNN and a probabilistic neural network based on a comparison sketch of various models; they discovered an architecture with both 3 × 3 and 7 × 7 kernels in an overlapped manner and built a cascaded architecture. Zhao et al. [17] proposed a method by integrating a fully convolutional neural network (FCNN) and conditional random fields (CRFs), rather than adopting CRFs as a post-processing step of the FCNN. However, a cascade of an FCNN was proposed by Wang et al. [18] to segment multi-modal MRI images with hierarchical regions: whole tumor, tumor core and enhancing tumor; the cascade is designed to decompose the multi-class segmentation problem into a sequence of three binary segmentations, the networks consist of multiple layers of anisotropic and dilated convolution filters, and they are combined with multi-view fusion to reduce false positives, while the proposed method by Zhao et al. [19] aims at segmenting image slices using deep learning models with integrated FCNNs and CRFs as recurrent neural networks from axial, coronal and sagittal views, respectively, while fusing segmentation results obtained in the three different views. On the other hand, Dong et al. [20] developed a novel 2D fully convoluted segmentation network that is based on the U-Net [21] architecture. In order to boost the segmentation accuracy, a comprehensive data augmentation technique was used in this work. In addition, they applied a ‘soft’ Dice-based loss function. Therefore, Sajid et al. [22] proposed a hybrid convolutional neural network (HCNN) architecture that uses a patch-based approach and takes both local and contextual information into account; when predicting the output label, the proposed network deals with an overfitting problem by utilizing a dropout regularizer alongside batch normalization. Meanwhile, Thaha et al. [23] developed an automatic segmentation method with skull stripping and image enhancement methods used in pre-processing and an HCNN used for segmentation with the loss function optimized by the Bat algorithm. Concerning 3D-CNN methods, Kamnistsas et al. [24] used an 11-layers-deep multi-scale 3D CNN; the architecture consisted of two parallel convolutional pathways that processed the input at multiple scales to achieve a large receptive field for the final classification while keeping the computational cost low. Mengqiao et al. [25] proposed an approach based on a 22-layers-deep three-dimensional convolutional neural network; they used several cascaded convolution layers with small kernels to build a deeper CNN architecture. 

Methods that use autoencoder architectures include that of Myronenko et al. [26] who proposed an encoder–decoder-based CNN architecture. They added an additional branch to the encoder endpoint to reconstruct the original image, similar to the autoencoder architecture; the motivation for using the autoencoder branch was to add additional guidance and regularization to the encoder part since the training dataset size is limited. They used the variational autoencoder approach to better cluster the features of the encoder endpoint. However, a novel architecture named residual cyclic unpaired encoder–decoder network (RescueNet) was proposed by Nema et al. [27] for brain tumor segmentation. They used training based on unpaired generative adversarial networks to train the RescueNet and a scale-invariant post-processing algorithm to enhance the accuracy.

Table 1 shows the performance results of the related works.

## 3. Proposed Method 

The proposed methodology in Figure 1 contains two main phases. The first is a training phase which concerns the preparation and augmentation of data followed by a CNN model. The second phase is the test phase in which we pre-process our test image, classify the image to find out if the tumor exists and finally segment the tumor according to the characteristics extracted from our model. These steps are discussed in detail in the following subsections.

### 3.1. Pre-Processing

We normalize all the images’ data by subtracting the mean μ of each input image i and dividing by the standard deviation σ to obtain $i_{0}$ pre-processing image as:(1)$$i_{0}=\frac{i-\mu }{\sigma }$$

### 3.2. Data Augmentation

Data augmentation is a method to reduce overfitting and generate more training data from the original data. In this article, we applied the enhancement methods summarized in Table 2 and simple transformations such as flipping rotation, noise addition and shifting. Figure 2 shows an example for all the transformations applied on the original image.

### 3.3. Convolution Neural Networks

Convolution neural networks (CNNs) have acquired a particular role in the last few years. They have proven to be very effective and are also most commonly used in various medical applications. They are composed of a succession of layers formed by neurons that carry out different operations on the input data. Convolution layers learn to link each image with its own category by detecting a number of representation feature maps *f*(*l*) at the layer *l* defined as:(2)$$f\left(l\right)=\sigma \left(f_{i}\left(l-1\right)*i_{l-1}+b_{l}\right)$$
where $f_{i}\left(l-1\right)$ is the feature maps of the earlier layer l−1; $i_{l-1}$ denotes the layer l−1; convolution operator is denoted by ∗; $b_{l}$ is a bias; and σ is a non-linear activation. The 2D structure of the image is taken into consideration, and it detects local features at various positions of the input maps. For each convolution layer, there is also a non-linear activation function (ReLU, sigmoid) defined as follows:(3)ReLU(x)=max(0,x)
(4)$$sigmoid\left(x\right)=\frac{1}{1+\exp\left(-x\right)}$$

Indeed, non-linearity implies that the output cannot be replicated using a linear combination from the inputs. The objective of pooling is to reduce the spatial dimension of representation feature maps through a variety of techniques such as average pooling and max pooling. Accordingly, it generalizes the outcomes of a convolution ensuring that the detection of characteristics is invariant when changing the scale or orientation. Lastly, fully connected layers are typically used in the final stages of the network, where features are transformed from high-level to low-level. However, this output can be flattened and connected to the output layer. The purpose of these final layers is to provide classifications concerning these features. CNNs are able to directly learn complex characteristics from the data. This is why brain research is focused on CNNs for tumor segmentation, where it is mainly focused on the conception of network architecture instead of image processing to extract their features.

There is still a scarcity of medical data in deep networks for medical research applications. The reason for this is the cost and complexity of the labeling of the images. To deal with the issue in the area of brain cancer imaging, we adopted a methodology to classify and segment the brain tumor using CNN.

#### 3.3.1. Tumor Classification (Tumor or Not Tumor)

Before segmenting the tumor, we should verify its existence; thus, we created a CNN model for binary classification with two neurons in the output layer.

The proposed CNN architecture was applied on MRI images; these types of images can show different tissue contrasts across different pulse sequences, making it an adaptable imaging technique widely used to visualize regions of interest in the human brain. There are four MRI sequences available for every patient: T1-weighted (T1), T1-weighted and contrast-enhanced (T1c), T2-weighted (T2) and T2-weighted FLAIR (FLAIR) (Figure 3). Each one of these modalities contains information that signifies performance improvement.

In our study, since our objective concerned complete tumor segmentation, we used the FLAIR image because it detects the tumor with peritumoral edema and it facilitates the segmentation of the whole tumor. 

Starting with classification, in this paper, we built an 18-layers-deep 2D CNN. The main architecture of the CNN is shown in Table 3.

The proposed CNN architecture, Figure 4, takes the FLAIR modalities as input in the first convolution layer, while succeeding layers take feature maps produced by the preceding layer as input; the network is composed of five convolution layers which, respectively, contain 512, 256, 128, 64 and 32 filters of the same size of 3 × 3, while each convolution works with a stride (the movement of the window over the input map) equal to 2 and a padding equal to SAME to make the output image have the same dimensions as the input image; at the end of each convolution, we applied the non-linear activation function ReLU.

The pooling layer comes after each convolution, in which we chose max pooling that selects the maximum value out of a specified window of size pl × pl to reduce the image parameters; then we chose a pool size of 2 × 2, stride equal to 2 and padding equal to SAME.

In the flattened layer, the previous layer output features maps are reshaped, and the neurons obtained as output are changed to obtain the one-dimensional vector; in our case, we obtained at the last layer of pooling a matrix of size 16 × 16 × 32 with 16 × 16 the size of feature map and 32 the number of these features, and this matrix was flattened in order to obtain features with a vector size 8192.

The fully connected (FC) layer contains 32 features that are used to predict the target class.

#### 3.3.2. Tumor Segmentation

After classifying the image and finding that the output prediction has a tumor, we should continue to the second task that is segmenting the tumor.

Most approaches to tumor segmentation depend on the ground truth that implies a subject intervention of the specialist. This problem is time-costuming and does not guarantee a 100% precise result. 

Our method solves these problems of labeled images and dispenses with the intervention of the specialist based on the features extracted from the CNN model.

The features extracted from the last convolution layer, Figure 5, are multiplied by a gradient according to these features in order to convey additional information that we assume detecting pixels that have more intensity and then segment the tumor area.

An image gradient quantifies a directional variation of intensity or color in an image. The gradient of the image is one of the fundamental building blocks in image processing; the term gradient or color gradient is also used for a gradual blend of color which can be considered as an even gradation from low to high values. Another name for this is color progression.

Calculate the gradient of an image; its mean calculates the variation of the intensity of each pixel, and the latter can be calculated as partial derivatives.

Let the function f(a,b) be the value of the image at position (a,b), then:(5)$$\frac{\partial f\left(a,b\right)}{\partial b}=\underset{\Delta \to 0}{lim}\frac{f\left(a,b+\Delta \right)-f\left(a,b\right)}{\Delta }\approx X\left[i,j+1\right]-X\left[i,j\right]=H\left[i,j\right]$$

If we take Δ=1, then we obtain an approximation X[i,j+1]−X[i,j]=H[i,j] which represents the horizontal change in intensity of a pixel at a certain position (i,j).

The same applies when calculating the vertical change of intensity by deriving the image *f* on a and fixing *b*; we then obtain:(6)$$\frac{\partial f\left(a,b\right)}{\partial a}=\underset{\Delta \to 0}{lim}\frac{f\left(a+\Delta ,b\right)-f\left(a,b\right)}{\Delta }\approx X\left[i+1,j\right]-X\left[i,j\right]=V\left[i,j\right]$$

If  H[i,j] and V[i,j] are the partial derivatives of the image *f*, then:

G[i,j,:]=[H[i,j],V[i,j]] is the gradient of image f in the position (i,j).

This method is similar to that proposed by R.R. Selvaraju [28] which used gradient-based localization for visual explanations. 

After generating all 32 images’ (features extracted from the last convolution layer) gradients, we searched their average and global maximum and pooled them in two vectors with size 32 instead of a matrix with shape (32,32,32) in order to obtain the neuron importance weights, minimize the features’ size and save time.

After that, we multiplied each feature map with corresponding pooled gradients, (component1 × filter1, component2 × filter2……..., component32 × filter32) stocked in the two vectors previously calculated, to extract more pertinent and detailed spatial information and specify the pixels that have a very high intensity which would be the tumor to segment. Then we obtained 32 significant images; since we needed just one, we calculated the mean image and thresholded it with a specific threshold to obtain our mask which precisely contained the tumor’s pixels. This mask was represented in grayscale intensities as it would be hard to measure small changes; then we applied a color map. The color map application allowed us to transmit the image size from (*N*, *N*) to (*N*, *N*, 3) which made it possible to calculate the addition between the original image and this map’s features to obtain a superimposed image and, finally, to have a well-segmented tumor; we ended with thresholding and post-processing to improve our segmentation.

### 3.4. Post-Processing

We post-processed the segmentation results by removing small regions around the corners of the MRI scan and correcting some pixels by a simple thresholding method and a morphological technique called opening.

## 4. Experimental Setup

### 4.1. Dataset

BraTS is a brain tumor image segmentation challenge. It is organized in conjunction with the International Conference on Medical Image Computing and Computer Assisted Intervention (MICCAI). Most of the state-of-the-art brain tumor segmentation methods have been evaluated on this benchmark.

The proposed method was tested and evaluated on the BraTS 2017 dataset; the training set contained 210 images of high-grade glioma (HGG) and 75 images of low-grade glioma (LGG) patient scans.

Multimodel MRI data were available for every patient in the BraTS 2017 dataset, and four MRI scanning sequences were performed for each patient using T1, T1c, T2 and FLAIR. The BraTS 2017 validation and testing set contained images from 46 and 146 patients with brain tumors of unknown grade, respectively.

For each patient, the T1, T2 and FLAIR images were coregistered into the T1c data, which had the finest spatial resolution, and then resampled and interpolated into $1\times 1\times 1\;{\mathrm{mm}}^{3}$ with an image size of 240 × 240 × 155.

The ground truth of training set was only obtained by manual segmentation results provided by experts.

### 4.2. Implementation Details

The algorithm was implemented in Keras library in Python. It is high-level library used for implementing neural networks and can run over either Theano or TensorFlow framework. It supports both GPU and CPU processing.

Hyper-parameters and many tools were tuned using grid search, and the parameters on which model performed best on validation data were selected. Firstly, to read the BraTS 2017 MRI images that had a NIfTI format type, we used SimpleITK, an open-source multi-dimensional image analysis in Python for image registration and segmentation.

Next, we chose FLAIR modalities for each image, cropping each image and saving their size as (192,152,3) instead of (240,240,3), the original size; we also chose the slice number 90 of 155 slices.

After image acquisition step and before training our CNN model, we augmented (Section 3.2) and pre-processed (Section 3.1) our data for better performance.

For the classification task, we assigned a ‘tumor’ or ‘not tumor’ label to each image based on the ground truth in order to simplify the segmentation task that comes afterward and avoid the case of segmenting images that had no tumor; then we had to focus on this part and try to eliminate all the false positive possible in order to have a good result during the segmentation.

The training dataset was divided randomly into training and testing sets with 70:30 ratios, the convolution layer kernels were initialized randomly with bias values set to zero, and the stride for all max pooling and convolution layers was set to two and, respectively, to produce translation-invariant feature maps. The best parameters for the proposed method are shown in Figure 4 and Table 3. The loss function used for our model was binary crossentropy by computing the following average:(7)$$Loss=-\frac{1}{outputsize}\overset{outputsize}{\underset{i}{\sum }}y_{i}\log{\hat{y}}_{i}+\left(1-y_{i}\right)\log\left(1-{\hat{y}}_{i}\right)$$
where ${\hat{y}}_{i}$ is the i-th scalar value in the model output, $y_{i}$ is the corresponding target value, and output size is the number of scalar values in the model output.

To minimize this loss function, we used Adam optimizer with initial learning rate of $\alpha_{0}=10^{-4}$ and progressively decreased it according to:(8)$$\alpha =\alpha_{0}\times {\left(1-\frac{e}{N_{e}}\right)}^{0.9}$$
where e is an epoch counter, and $N_{e}$ is the total number of epochs; in our case, the maximum number of epochs = 45, and in every epoch, the batch size = 20.

In the segmentation part, precisely in the post-processing step shown in Figure 6, we configured some parameters in thresholding and opening steps.

Since the choice of threshold was not fixed for each image because of the variation of the pixel’s intensities, we decided to choose several thresholds and test the performance of each threshold by calculating the similarity coefficient of each segment that tumor receives from this current threshold and the ground truth in order to choose the best threshold.

We tested more than 22 thresholds for each grayscale image ($\tau_{1}=0.33,\tau_{2}=0.35,\tau_{3}=0.37,\ldots \ldots \tau_{21}=0.83,\tau_{22}=0.85$).

For the opening stage, we used a small kernel 3 × 3 to delete some insignificant pixels without effect on the tumor.

### 4.3. Performance Evaluation

The experimental results were evaluated using different types of performance indicators: precision, recall and accuracy for classification task and Dice similarity coefficient (DSC) for segmentation task:Precision: It is the percentage of results that are relevant and is defined as:
$$Precesion=\frac{TruePositive}{TruePositive+FalsePositive}$$Recall: The percentage of total relevant results correctly classified by the proposed algorithm which is defined as:
$$Recall=\frac{TruePositive}{TruePositive+FalseNegative}$$Accuracy: Formally, accuracy has the following definition:
$$Accuracy=\frac{TruePositive+TrueNegative}{Total}$$The DSC represents the overlapping of predicted segmentation with the manually segmented output label and is computed as:
$$DSC=2\times \frac{\left|G\cap S\right|}{\left|G\right|+\left|S\right|}$$
where *G* and *S* stand for output label and predicted segmentation, respectively.

## 5. Results and Discussion

In this section, we present the results concerning the two tasks of binary classification of tumor or non-tumor and segmentation of the complete tumor.

### 5.1. Binary Classification of Tumor or Non-Tumor

In this part, we used the BraTS 2017 training set by mixing two types of gliomas, HGG and LGG, with data augmentation to improve the classification task. The set contained 1100 labeled images, 550 for each label. This set was divided into two subsets for training and validation (70:30 ratios). Figure 7 shows the performance of the method proposed: 98% and 91% accuracy for the training and validation subsets, respectively.

Table 4 and Figure 8 show the other performance evaluation indicators and the ROC curve, respectively. The evaluation results of binary classification achieved good performance, and the ROC curve showed good classification close to perfect classification for both tumor and non-tumor labels. This high classification performance can facilitate the second task of segmentation by eliminating the case of segmenting patients that do not have a tumor.

### 5.2. Tumor Segmentation

After classifying the image as a tumor, segmentation is the next step. The evaluation of the segmentation method was applied on the BraTS 2017 training dataset based on the dice similarity coefficient between the true label and the prediction.

Figure 8 shows that our ROC curve is only close to the perfect ROC curve, which means that our binary classification is high performance. 

Table 5 shows the Dice scores of our method before and after post-processing of HGG gliomas, LGG gliomas and mixing the two types HGG and LGG.

Our method achieved 83.59%, 79.59% and 82.35% DSCs in the BraTS 2017 training dataset for HGG, LGG and mixing HGG and LGG patients, respectively.

Table 5 shows that the post-processing was very impressive in terms of increasing the value of the DSC by removing small regions around the corners of the MRI scan and correcting some pixels that belong to the non-tumor area.

In Table 6, it is shown that our method does not have the best performance, but we can say that it is good and acceptable. The methods [18,27,29] of tumor segmentation use the ground truth in their algorithms to obtain a high-performance result and during evaluation. On the other hand, the methods [30,31,32] are shallow unsupervised methods for tumor segmentation without using the ground truth; our method belongs to these unsupervised ones based on the features extracted from the CNN and the post-processing to reach our objective of tumor segmentation. 

From Table 6, we can say that our method is better than the unsupervised methods that exist and is of the same performance as supervised methods based on deep learning.

Qualitative segmentation results are presented in Figure 9 and Figure 10 for the BraTS 2017 dataset. The figures also indicate that our network is capable of accurately segmenting complete tumor regions. Our approach was tested on the FLAIR modalities for MRI images outside the BraTS 2017 dataset, and it provided good results, which guarantees the performance and power of the proposed method.

## 6. Conclusions

In this article, we proposed an approach based on CNN architecture in order to predict and segment simultaneously a cerebral tumor. In this process, an MRI image was pre-processed and augmented using normalization and data augmentation techniques. The MRI image was classified into a tumor or not tumor brain image by a CNN model with two neurons in the output layer; in this task, we used the ground truth to label the images as tumor or not tumor images. The segmentation was applied on the images that contained the tumor, using the features extracted from the last convolution layer of our CNN architecture and their gradients. Finally, we applied post-processing to improve our results.

The strength of our approach is demonstrated by dispensing with the intervention of a specialist in order to manually locate the tumor pixel by pixel because it is a complex and time-consuming task.

Our method solves these problems by using the features extracted from the CNN architecture and independently of the ground truth detected manually by specialists. The experimental results show good performance and a significant result when compared with the existing methods. The compared parameters were precision, recall and accuracy for the binary classification and the Dice coefficient score in the segmentation task.

Future work will be devoted to improving these results and using deeper architectures to increase the performance of the segmentation output.

## Figures and Tables

**Figure 1 jimaging-07-00269-f001:**
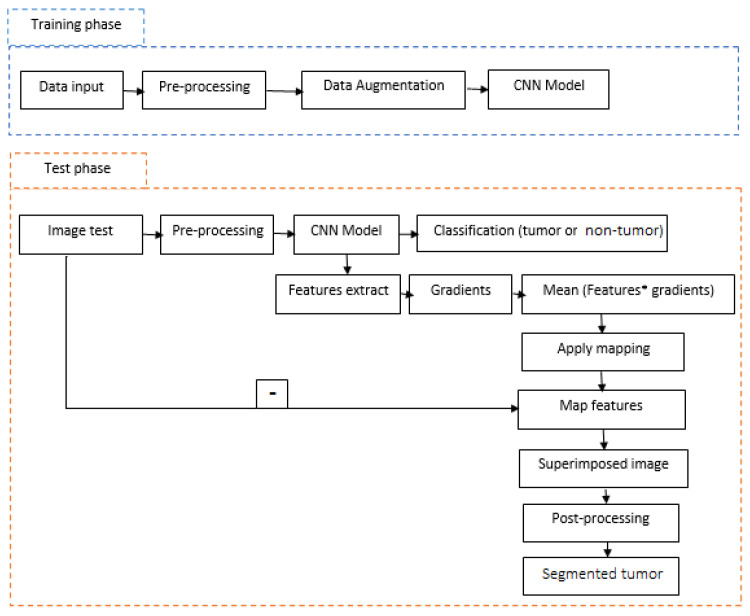
General architecture of the proposed method.

**Figure 2 jimaging-07-00269-f002:**
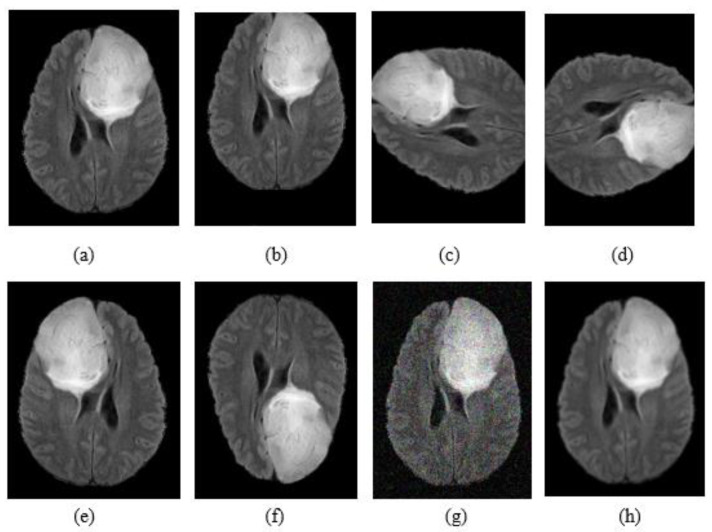
Example of data augmentation: (**a**) FLAIR image; (**b**) shift; (**c**) rotation +90°; (**d**) rotation -90°. (**e**) Flip horizontally; (**f**) flip vertically; (**g**) noise addition; (**h**) blur.

**Figure 3 jimaging-07-00269-f003:**
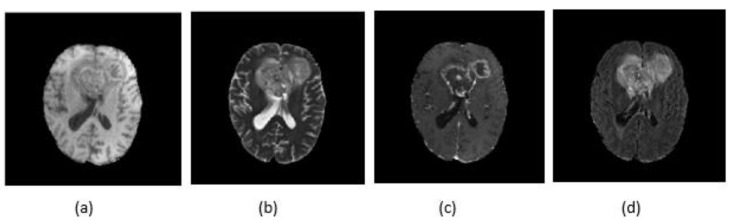
Different modalities of MRI images: (**a**) T1; (**b**) T2; (**c**) T1c; and (**d**) FLAIR.

**Figure 4 jimaging-07-00269-f004:**
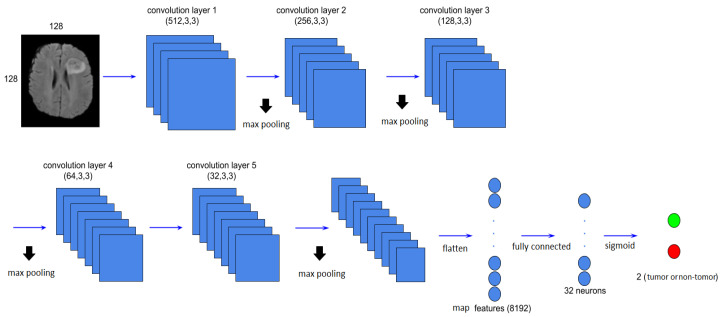
CNN architecture used in our approach.

**Figure 5 jimaging-07-00269-f005:**
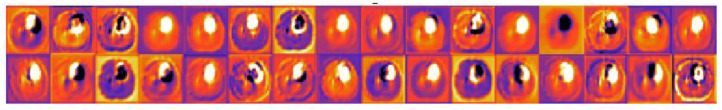
Features extracted from the last convolution layer of our CNN model.

**Figure 6 jimaging-07-00269-f006:**
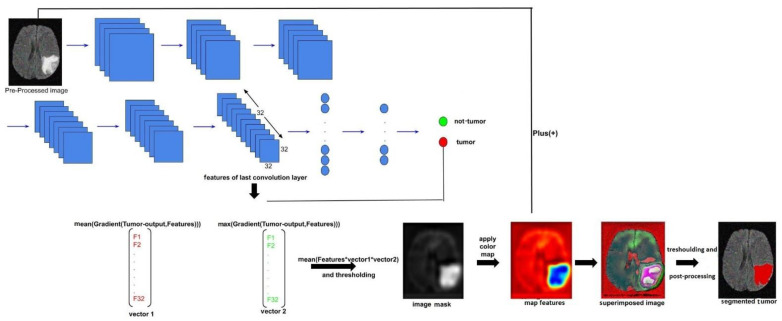
Illustration of our tumor segmentation process of an MRI image from training BraTS 2017 dataset.

**Figure 7 jimaging-07-00269-f007:**
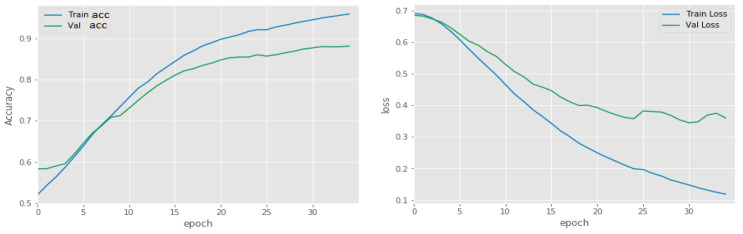
Accuracy and loss of the training and validation subsets as a function of number of epochs.

**Figure 8 jimaging-07-00269-f008:**
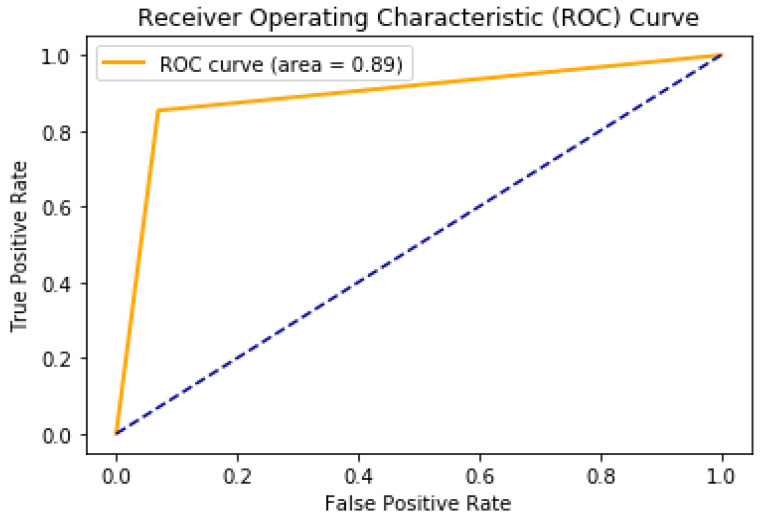
ROC curve of our binary classification model to detect the existence of the tumor.

**Figure 9 jimaging-07-00269-f009:**
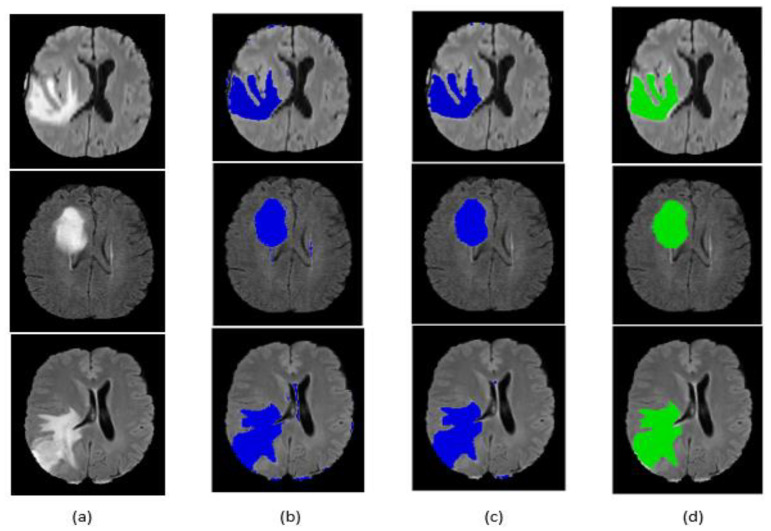
Segmentation result of our method on some BraTS 2017 HGG images: (**a**) original image, (**b**) segmentation before post-processing, (**c**) segmentation after post-processing, (**d**) ground truth.

**Figure 10 jimaging-07-00269-f010:**
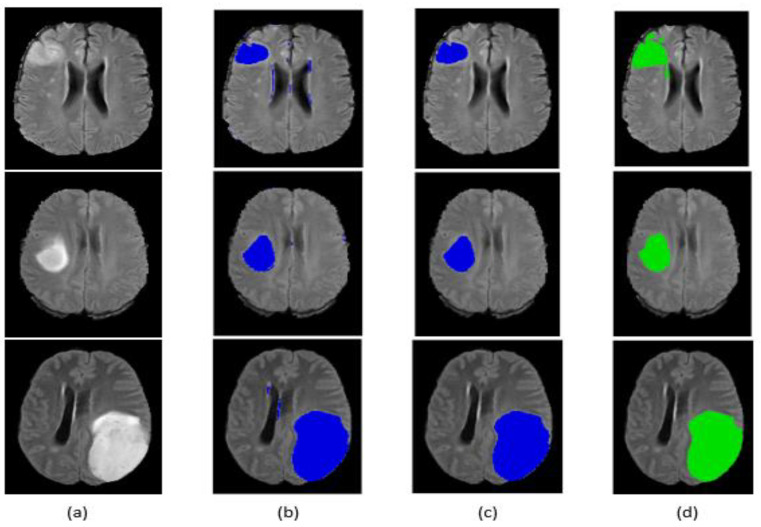
Segmentation result of our method on some BraTS 2017 LGG images: (**a**) original image, (**b**) segmentation before post-processing, (**c**) segmentation after post-processing, (**d**) ground truth.

**Table 1 jimaging-07-00269-t001:** Summary of related works’ performance results.

Method	Dataset	Dice Similarity Coefficient (Whole)
Lyksborg et al. [13]	BraTS 2014	79.9%
Pereira et al. [14]	BraTS 2013	84%
Havaei et al. [15]	BraTS 2012	82%
Wang et al. [18]	BraTS 2017	87%
Zhao et al. [19]	BraTS 2012	80%
Dong et al. [20]	BraTS 2015	86%
Kamnistsas et al. [24]	BraTS 2016	85%
Myronenko et al. [26]	BraTS 2018	81%
Nema et al. [27]	BraTS 2018	94%

**Table 2 jimaging-07-00269-t002:** Summary of the applied data augmentation methods.

Methods	Range
Flip horizontally	50% probability
Flip vertically	50% probability
Rotation	±90° degree
Shift	20 pixels in horizontal and vertical direction
Noise addition	Random noisy
Blur image	Gaussian blur

**Table 3 jimaging-07-00269-t003:** The main architecture of CNN.

	Type	Filter Size	Stride	# Filters	FC Units	Output
Layer 1	Conv	3 × 3	1 × 1	512	-	512 × 128 × 128
Layer 2	Activation	-	-	-	-	512 × 128 × 128
Layer 3	Max pool	2 × 2	2 × 2	-	-	512 × 64 × 64
Layer 4	Conv	3 × 3	1 × 1	256	-	256 × 64 × 64
Layer 5	Activation	-	-	-	-	256 × 64 × 64
Layer 6	Conv	3 × 3	1 × 1	128	-	128 × 64 × 64
Layer 7	Activation	-	-	-	-	128 × 64 × 64
Layer 8	Max pool	2 × 2	2 × 2	-	-	128 × 32 × 32
Layer 9	Conv	3 × 3	1 × 1	64	-	64 × 32 × 32
Layer 10	Activation	-	-	-	-	64 × 32 × 32
Layer 11	Conv	3 × 3	1 × 1	32	-	32 × 32 × 32
Layer 12	Activation	-	-	-	-	32 × 32 × 32
Layer 13	Max pool	2 × 2	2 × 2	-	-	32 × 16 × 16
Layer 14	Flatten	-	-	-	8192	-
Layer 15	FC	-	-	-	32	-
Layer 16	Activation	-	-	-	32	-
Layer 17	FC	-	-	-	2	-
Layer 18	Activation	-	-	-	2	-

“#” means “the number of” and “- “means “non”.

**Table 4 jimaging-07-00269-t004:** Tumor binary classification: precision, recall and accuracy of the training BraTS 2017 dataset.

	Training Subset (HGG + LGG)	Validation Subset (HGG + LGG)
Precision (%)	99	92
Recall (%)	99	91
Accuracy (%)	98	91

**Table 5 jimaging-07-00269-t005:** The Dice scores of our segmentation method before and after post-processing on the BraTS 2017 training dataset.

	Training Dataset (HGG)	Training Dataset (LGG)	HGG + LGG
Befor post-processing	77.66%	74.6%	76.88%
After post-processing	83.59%	79.59%	82.35%

**Table 6 jimaging-07-00269-t006:** Performance comparison between our proposed method and different CNN approaches on BraTS 2017 dataset.

Methods	Data	Performance of Complete Tumor
Single-Path MLDeepMedic [29]	BraTS 2017	DSC 79.73%
U-NET	BraTS 2017	DSC 80%
RescueNet [27]	BraTS 2017	DSC 95%
Cascaded Anisotropic CNNs [18]	BraTS 2017	DSC 87%
Force Clustring [30]	BraTS	-
K-means and FCM [31]	https://radiopaedia.org/ (accessed on 3 May 2021)	ACC 56.4%
Bi-secting (No Initialization) [32]	MRI images collected by authors	ACC 83.05%
**Proposed**	**BraTS 2017**	**82.35%**

## Data Availability

We use an open data from Kaggle BraTS2017. The link: https://www.kaggle.com/xxc025/unet-datasets (accessed on 1 January 2020).

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
