# Peer review of "Brain Tumor Segmentation Based on Deep Learning’s Feature Representation"

_2313-433X, 2021, doi:10.3390/jimaging7120269_

Round 1

Reviewer 1 Report

The paper presents an interesting subject. The following aspects must be added in order to increase the soundness of the paper:

  • section related work must be divided in methods for classification and for segmentation;
  • section related work must contain also results of the existing methods; a comparison of the existing methods must be done; 
  • explanation of the used architecture must be added; why are the benefits of the proposed architecture for both classification and segmentation
  • how do you explain the better results obtained in papers 31 and 32?
  • fig 1, page 7 must be renumbered

Author Response

First of all, we would like to express our thanks to the learned editor and two reviewers for their constructive and interesting suggestions, which improved the quality of our paper.

Reviewer 2 Report

Title: Brain tumour segmentation based on deep learning's feature representation

Summary:

This works presents a segmentation method for brain tumour in FLAIR images that uses tumour identification as a pre-training task for obtaining saliency maps, that are employed lastly to generate the final tumour delineation.

The manuscript is scientifically sound but presents several limitations with respect to readability and needs a thorough review of the writing. Furthermore, the method is not properly emphasized in my opinion as a segmentation method that does not need ground truth segmentations.

  1. The text presents multiple errors at the moment with respect to the writing. See some errors below:

    “compliacated”, “machin”, “segmantation”, “ti the subregion”, “multy-view”, “is adaptive to”, “[...] small kernels to building”, “
    developped”, “convuluyion”

    Additionally,
    there is a list of varied concerns about the writing:

    1. When defining terms that will have a corresponding acronym, the text is all written in lowercase (unless the word is always written with a capital letter, such as a person’s name). Therefore, “magnetic resonance imaging (MRI)” is correct, but “Discrete Wavelet Transform (DWT)” should be all in lowercase. This is repeated throughout the text. See acronyms ANN, KNN, FCNN or CNN.

    2. Also, some acronyms are never used again in the manuscript, such as HCNN or VAE. Please, remove acronyms that are only used once.

    3. When citing Menze et al. [7], the authors said “a healthy brain tumour atlas and a latent brain tumour atlas combined [...]”. I believed there is a verb (“were combined”) missing.

    4. Also, when citing the literature, it is enough to cite the surname. The name initials are not necessary. So “B.H. Menze” is “Menze”.

    5. “manually made features” should be “manually defined features”.

    6. Fifth paragraph of the “related work” section: “training procedure that allow us” should be “training procedure that allow the authors”.

    7. Several acronyms appear in the introduction and were never defined (for example, CNN or MR). These acronyms are later defined in the second section and CNN is defined up to four times. Please, define each acronym once and when it first appears.

    8. Fifth paragraph of “related work” section: “an encoder-decoder approach-based CNN” probably means “an enconder-decoder-based CNN”?

    9. RNN is never defined in the manuscript. BAT is never defined. GAN is never defined.

  1. The introduction is a bit incoherent. A better connection is needed to understand the state-of-the-art of the problem that this work is trying to solve. See some concerns about it:

    1. The first sentence about machine learning methods applied to recognize the brain tumour only presents the methods and nothing is commented about the input features used (despite being a crucial part in this task).

    2. The text explaining ref. [6] is not clear. More details are needed about the method. Are they using this probability map for segmentation, for classification or both?

    3. In ref. [8], what did the authors used as input for the fuzzy SVM? Other references have details about feature extraction, but this one and the ones mentioned before do not.

    4. When presenting the proposed method, the authors give too much technical details about their method and this hinders the understanding. I advise the authors to simplify this first presentation of their work and explain it clearly so that a reader can understand it without reading the method section.

  1. The “related work” section is difficult to follow. Some paragraphs are split in the middle of a presentation of some work (see for example end of fourth paragraph and the beginning of the fifth). I advise the authors to present it in a more comprehensive way. An option is to group the related literature by type of approach (for example, works that used autoencoders and works that used convolutional networks).

  2. In the “related work” section (fifth paragraph), the authors write “a convolutional implementation of a fully connected layer”. Could the authors explain what does this sentence mean? Is it a complicated way of referring to a convolutional layer?

  3. In the same paragraph as in the previous point. What do the authors mean by “a cascaded convolutional layer”? Usually, the term cascaded is used for networks that are concatenated so that the output of the previous one is the input for the next one. The meaning is not clear when talking about layers. Please, explain it further.

  4. In last paragraph of “related work” section. The authors write “an automatic segmentation method such as Skull stripping and image enhancement methods”, while probably they mean that this “automatic method” uses “skull stripping and image enhancement”, right? These methods are not segmentation methods.

  5. Same paragraph as in previous point. The authors write “with both 3x3 and 7x7 in an overlapped manner”, but I believe that the world “kernel” is missing. Otherwise, the sentence is incomplete.

  6. Section 3.1. is explained in a complicated manner and difficult to understand. The pre-processing is a basic normalization to a mean 0 and standard deviation 1 distribution.

  7. Section 3.2. I guess that “add noisy” is “noise addition”.

  8. In Table 1, could the authors provide the actual range of rotations? For example, (-20,20) degrees. Could the authors provide also the other parameters used in every data augmentation method when available? This is important for ensuring that the study can be replicated.

  9. The tumour segmentation method is not very clear at the moment. I understand that after extracting image horizontal and vertical gradients for 32 feature channels, the mean and maximum values are extracted. However, little detail is given for how these features are translated to original image size. In summary, how are the two 32-dimensional vectors used to extract the saliency maps shown in Figure 6 with image dimensions 128x128? Also, what is the final threshold used out of the 22 that were tested? Please, provide more details.

  10. Overall, I think that the method and results should emphasize that the proposed methods segments brain images without actual groundtruth, only by identifying brain tumours in each slice as a pre-training task. A literature review on unsupervised methods or methods that perform segmentation indirectly is suggested and a comparison of accuracy against these type of methods also (together with results in Table 5).

Author Response

(The authors gave the same response as above.)

Round 2

Reviewer 1 Report

Since all my comments were addressed, I recommend to publish the paper.

Reviewer 2 Report

The authors answered all my concerns. I have minor comments to further improve the text, especially with regards to the writing.

1. See some errors introduced together with the new text:

“mchines”, “Laplacien”, “gradiant”, “gaussiens”, “flter”, “ositives”, “composent”, “Howover”, “composant”, “gradien”, “iintensity”, “SIMPLTIK” is “SIMPLEITK”?, “Conclusioin”

I really recommend to use a spell-checking software to identify these errors.

2. Additionally, there are sentences that are too long and difficult to follow. I would rather split them in small sentences with clear meanings. An example is “In addition, Havai et al [...], their networks [...], they also described [...]”.

3. I would include also the references to the bibliography in the newly introduced Table 1.

4. Figure 4 is repeated now or is it just the version control? Please, verify. Same for Figure 5.

5. In Table 5, what does Random mean for the last method?
